# Outcomes and complications among nonagenarians undergoing cardiac surgery: A scoping review

Laurence Weinberg[1,2]*, Jarryd Ludski[1], Bradly Carp[1], Je Min Suh[1], Anoop N. Koshy[3,4], Cilla Haywood[2,5], Benjamin Churilov[1], Dong-Kyu Lee[6], Michael Yii[7]

**1** Department of Anaesthesia, Austin Health, Heidelberg, Australia, **2** Department of Critical Care, The University of Melbourne, Parkville, Victoria, Australia, **3** Department of Cardiology, Austin Health, Melbourne, Victoria, Australia, **4** Department of Medicine, The University of Melbourne, Parkville, Victoria, Australia, **5** Department of Aged Care, Austin Health, Heidelberg Heights, Victoria, Australia, **6** Department of Anaesthesiology and Pain Medicine, Dongguk University Ilsan Hospital,Goyang, the Republic of Korea, **7** Department of Cardiac Surgery, Epworth Eastern Hospital, Box Hill, Victoria, Australia

* laurence.weinberg@austin.org.au

## Abstract

### Introduction

This review was aimed at understanding the scope of evidence regarding outcomes and complications in nonagenarians (90–99 years of age) undergoing open cardiac surgery.

### Methods

The review was conducted in accordance with the Preferred Reporting Items for Systematic Reviews and Meta-Analyses (PRISMA) Extension for Scoping Review Protocol guidelines. A search of three databases, MEDLINE, EMBASE, and the Cochrane Central Register of Controlled Trials, identified articles pertaining to nonagenarians undergoing various open cardiac surgical procedures. No restrictions were applied to study design or publication date.

### Results

From the initial 1826 articles identified, we included 28 studies from eight countries in a total of 6411 nonagenarians. The median 30-day mortality rate was 10.5% (IQR 7.2–14.6). Postoperative complication rates were reported in 20 studies (71%), and the median major complication rate was 71.4%. Respiratory, cardiac, renal, neurologic, gastrointestinal, and/or infectious complications were reported in 19%, 20%, 14%, 18%, 5%, and 9% of cases, respectively. The median length of hospital stay was 12.5 days (IQR 10.4–18.0). No studies reported unplanned readmissions to the intensive care unit or detailed patient-centered outcome measures.

**Data availability statement:** All relevant data are within the paper and its Supporting information files.

**Funding:** The author(s) received no specific funding for this work.

**Competing interests:** The authors have declared that no competing interests exist.

## Conclusions

Although age alone should not preclude nonagenarians from undergoing cardiac surgery, the procedure is associated with a significantly elevated risk of morbidity and a relatively high mortality rate. The review findings emphasize the need for international registry data to identify risk factors associated with adverse outcomes, explore strategies to decrease the risk of major complications, and improve postoperative quality of life. Moreover, creating and implementing uniform preoperative frailty assessments, and correlating them with surgical outcomes, will be crucial. Developing standardized patient-reported experience and outcome measures will also be imperative. Scoping review registered on OSF registries (https://osf.io/4mg7n).

## Introduction

Global life expectancy has markedly increased. In many countries, the life expectancy exceeds 85 years, particularly among women [1]. Consequently, the expanding population of nonagenarians, individuals 90–99 years of age, is the fastest-growing demographic within the older population [2]. This demographic transformation underscores the growing importance of addressing the unique healthcare needs and challenges faced by this rapidly expanding age group. Historically, cardiac surgical options for nonagenarians have been limited; however, advancements in surgical techniques and anesthesia have increasingly enabled nonagenarians to safely undergo such procedures [3].

Given that ischemic and valvular heart disease, and acute coronary syndromes are highly prevalent among older people [4–8], the number of nonagenarians expected to undergo cardiac surgery is anticipated to rise. Because ischemic and valvular heart disease remains prevalent in people ≥90 years of age [4–8], and the nonagenarian population is projected to exceed 30 million people by 2030 [2], a clear and yet underexplored gap exists in age-specific perioperative outcome data. Nonagenarians often present with multiple comorbidities that decrease their functional reserves, thereby heightening their susceptibility to postoperative complications and increasing their mortality risk [9,10]. Despite the well-documented rise in the incidence of cardiovascular disease among nonagenarians and the increasing number of patients undergoing cardiac surgeries [3,11], data specific to this demographic and adequate information on age-specific outcomes remain lacking. Understanding these parameters will be crucial for guiding future research and equipping healthcare providers with the resources necessary to meet the surgical needs of nonagenarians. These findings might also substantially affect the allocation of healthcare resources. Therefore, we conducted a scoping review to identify critical knowledge gaps in the literature, to provide a foundation for guiding research aimed at advancing the understanding of nonagenarians undergoing cardiac surgery.

## Objectives

This review provides an overview of the current landscape of cardiac surgery in nonagenarians, detailing inpatient, 30-day, and long-term complication and mortality rates. Additionally, the review evaluates key metrics, such as the lengths of intensive care unit (ICU) and hospital stays, as well as the frequency of inpatient and hospital readmissions. Finally, as an exploratory outcome, the review identified whether patient-reported outcomes (PROMs) and experience measures (PREMs) have been included in studies examining outcomes in nonagenarians undergoing cardiac surgery, given that an absence of such data would itself constitute a notable finding. By addressing these objectives, this review highlights critical aspects of perioperative care and identifies gaps in the existing literature, to better inform future clinical practice and research.

## Materials and methods

### Study methods

This review was conducted to examine the scope of the existing literature on the outcomes and complications associated with cardiac surgery in nonagenarians. Anticipating a dearth of comprehensive studies, we used a broad search encompassing both peer-reviewed and grey literature mentioning "nonagenarians," "cardiac surgery," and related terms. Grey literature refers to materials produced outside commercial publishing and traditional academic frameworks, including institutional reports (e.g., government-issued policy documents), conference materials (e.g., proceedings and abstracts), and organizational publications (e.g., professional association guidelines or collaborative research group working papers).

This review was conducted according to the Preferred Reporting Items for Systematic Reviews and Meta-Analyses (PRISMA) Extension for Scoping Reviews (PRISMA-ScR) guidelines (S1 Table), to ensure methodological rigor and transparency throughout the review process [12]. We systematically reviewed the existing literature and applied the methodological frameworks of Arksey and O'Malley [13] and Levac et al. [14]. These methods have been described in detail in the scoping review protocol [15].

### Protocol and registration

The study protocol was developed in collaboration with senior perioperative physicians specializing in cardiac surgery, to ensure a robust and clinically relevant approach. The protocol was designed to prioritize comprehensive evaluation of clinical outcomes following cardiac surgery in nonagenarians, addressing the unique challenges and considerations associated with this rapidly growing demographic. The protocol underwent peer review and was published in BMJ Open [15]. This process ensured effective protocol dissemination, public accessibility, and transparency. The review was also registered on Open Science Framework registries (https://osf.io/4mg7n).

### Ethical considerations

This study relied on secondary data analysis, which is available in database of scientific literature and, therefore, it did not require submission to the Austin Health Human Research Ethics Committee. All analysis of data were conducted in accordance with the ethical standards of the research committee and with the 1964 Declaration of Helsinki and its later amendments.

### Search strategy

A search of three databases, MEDLINE, EMBASE, and the Cochrane Central Register of Controlled Trials, identified articles pertaining to nonagenarians undergoing various open cardiac surgical procedures. To ensure comprehensive literature coverage, we identified supplementary studies through manual tracking of the references in the included articles. The search strategy is presented in S2 Table.

## Types of studies

Studies and abstracts deemed eligible for inclusion in this review are summarized in Table 1. Editorials, study protocols, and dissertations were excluded from consideration, to maintain the focus on original research and ensure methodological rigor.

## Eligibility criteria

This review incorporated publications (with no start date restriction, published until August 29, 2023) reporting data on patients 90–99 years of age. The included literature comprised primary empirical studies, randomized controlled trials, cross-sectional studies, cohort studies, and case reports, alongside full-text conference proceedings, abstracts, and posters. Non-English manuscripts were excluded unless they were accompanied by an English-language abstract meeting the eligibility criteria, in which case the full text was translated for evaluation. Restricting the search to English-language sources, despite potentially introducing geographic bias and limiting the generalizability of the findings, ensured methodological consistency in analysis and interpretation. This approach aligned with the exploratory nature of this review, which was aimed at systematically mapping available evidence rather than synthesizing global findings. Notably, a recent review supports that such language restrictions have minimal influence on the effect estimates or conclusions in most cases [16].

Given that the key aim of this review was to provide a synopsis of cardiac surgery in nonagenarians, the eligibility criteria included a wide range of open cardiac surgical procedures, as reported in the scoping review protocol [15]. These procedures encompassed coronary artery bypass grafting (CABG) and any valvular surgery (e.g., aortic, mitral, tricuspid, or pulmonary valve repair or replacement), atrial or ventricular septal defect closures; minimally invasive valvular surgeries requiring cardiopulmonary bypass, surgeries on the ascending aorta or aortic arch via sternotomy or thoracotomy, including hybrid stent-graft prostheses such as the frozen elephant Trunk procedure; heart transplantation; ventricular assist device implantation; left ventricular aneurysmectomy; surgeries on the descending aorta via sternotomy or thoracotomy (e.g., thoracoabdominal aortic repair or replacement); and removal of cardiac tumors or masses, with or without cardiopulmonary bypass.

Because they did not involve open-heart surgery, transcatheter aortic valve replacement and transcatheter aortic valve implantation procedures were excluded from this review. We also excluded any percutaneous or transapical valvular implants or interventions, e.g., mitral valve clip implantation. Finally, open and endovascular abdominal aortic repairs, as well as thoracic endovascular aortic repairs, were excluded. These exclusions ensured a focus solely on the outcomes of open-heart surgical interventions. Further details regarding these procedures can be found in the publications selected for review [17–44] and their respective reference lists.

**Table 1. Inclusion and exclusion criteria.**

|  | Inclusion | Exclusion |
|---|---|---|
| Population | • Participants 90–99 years of age (nonagenarians) | • Participants younger than 90 years or older than 100 years |
| Concept | • Studies evaluating complications, mortality, or any clinical outcomes among nonagenarians undergoing cardiac surgery<br>• Studies evaluating psychosocial or behavioral outcomes | • Studies focused on non-health outcomes, e.g., health economic evaluations |
| Context | • Assessment of perioperative outcomes in nonagenarians undergoing open cardiac surgical interventions |  |
| Types of evidence | • Primary empirical research studies (e.g., randomized controlled trials, cohort studies, cross-sectional studies, or case reports)<br>• Full-text articles, including abstracts, written in English<br>• Published abstracts or posters<br>• Grey literature, including published government reports and policy documents, conference proceedings and abstracts, professional association publications, and published working papers from research groups or committees | • Editorial articles (e.g., perspective pieces or position statements)<br>• Study protocols<br>• Dissertations<br>• Clinical trial registries<br>• Preprints and non-peer-reviewed research reports |

## Study selection and screening procedure

A two-step screening process was conducted with the Covidence® web-based review platform. Our search identified eligible studies, and duplicates were excluded. A pilot test using our predefined eligibility criteria was then conducted on 200 randomly selected articles, to ensure the robustness of the data collection instruments before the full study. This test, performed by two authors (JL and BC), was also used to calibrate the reviewers' use of screening protocols and ensure uniform application of the selection criteria.

The pilot review process led to the inclusion of nine studies and the exclusion of 191 studies, thus demonstrating the rigor of the selection method. Reviewer agreement was high, with a 95.6% concordance rate and a Cohen's kappa coefficient of 0.643, indicating substantial inter-rater reliability. Full-text publications of all relevant and potentially eligible studies were retrieved and independently screened by two reviewers (JL and BC). Discrepancies were resolved through adjudication by a third reviewer (LW). Studies not meeting the inclusion criteria were excluded, and the entire process was systematically documented in adherence with PRISMA guidelines to ensure transparency and reproducibility.

## Data extraction

The included studies were systematically charted in a customized data extraction form to ensure comprehensive, standardized collection of all relevant information, as previously reported [15]. Postoperative outcomes were meticulously captured, including complications, inpatient mortality, and mortality rates at 30 days, 1 year, 5 years, and 10 years. Additionally, PROMs and PREMs were extracted to assess long-term effects from the patient perspective. This systematic approach ensured a robust dataset for analysis and synthesis.

## Data synthesis and analysis

Data analysis was conducted in GraphPad Prism (v10.4.1, GraphPad Software, San Diego, CA, USA). Continuous variables were assessed for normality with the Kolmogorov–Smirnov test and/or Q-Q plot visual inspection. Normally distributed data are presented as mean ± standard deviation, whereas nonparametric data are summarized as median (interquartile range [IQR]) with full range (minimum–maximum). Categorical variables were analyzed with the chi-square test or Fisher's exact test, depending on the expected cell frequencies. For studies with substantial heterogeneity, a narrative-synthesis approach was used to contextualize patient-reported outcomes. A comparative analysis of study designs and participant characteristics was performed to identify research gaps.

This process, through systematically examining variations in methods and demographic profiles across studies, highlighted areas in which the existing evidence is limited or inconsistent, to potentially guide future research priorities and enhance understanding of the investigated topic. Our approach can therefore be accurately described as a narrative synthesis of quantitative findings, organized thematically around clinical domains (mortality, morbidity, and length of stay). This methodological flexibility enabled nuanced exploration of the complexity present in the literature while maintaining adherence to a coherent analytical framework. This framework facilitated the identification of knowledge gaps and the recognition of emerging trends within the field.

## Patient and public involvement

This scoping review exclusively analyzed existing research and did not involve members of the public or direct participation from patients in its design, conduct, or dissemination.

## Results

### Study selection

From the initial screening process, a total of 1826 articles were identified. An additional 20 articles were identified through a search of the grey literature and reference tracking, to supplement the primary database search. However, no eligible

articles were identified from the grey literature. In total, 81 full-text publications were assessed for eligibility, among which 28 studies [17–44] met the inclusion criteria and were included in the final analysis. The PRISMA flow diagram detailing the selection of studies is presented in Fig 1.

Although a narrative synthesis approach was used for quantitative data, no qualitative themes were identified. In addition, our findings are descriptive and were derived from unweighted medians rather than means or pooled statistics, to avoid overinterpretation.

## Study characteristics

All studies included in our review were reported in English (abstract and/or full text) and published between 1994 and 2022. The findings reported in these studies were collected between 1983 and 2019, during a median collection period of 9 years. All studies were retrospective in design. Whereas 18 studies (64%) were performed in the United States [17–22,25,26,28,34–37,39–43], the others were performed in Canada [22], the United Kingdom [23,27], Germany [31,32], Saudi Arabia [29], Australia [44], France [24], Japan [38], and Italy [30]. A total of 19 studies (68%) were performed in public hospitals [17–27,31–34,37–39,41], four (14%) were performed in private hospitals [30,35,40,42], and three (10%) were performed in both public and private facilities [28,36,44]. Study characteristics and outcome measures are presented in Table 2.

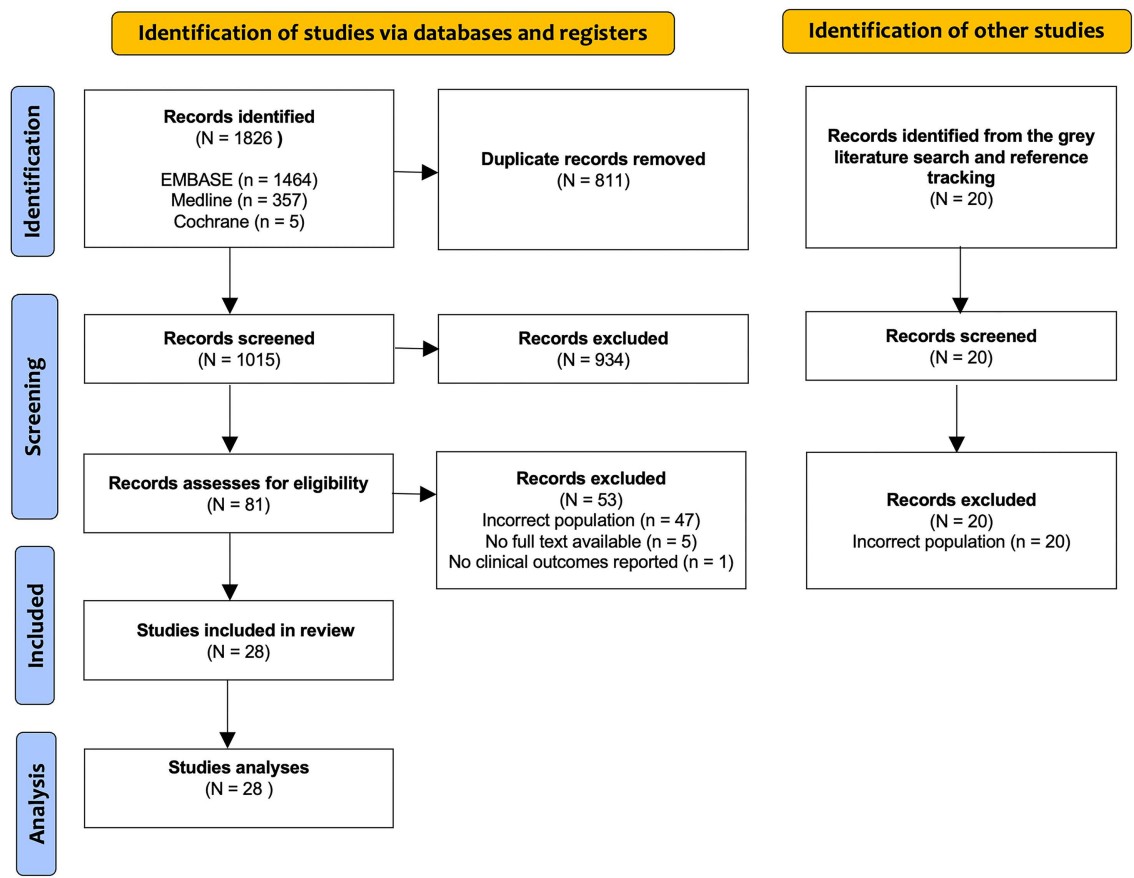

**Fig 1. Study selection flow diagram.**

**Table 2. Key characteristics of the included publications.**

| Authors | Year of publication | Data collection period | Country | Institution | No. of patients | Male | Female |
|---|---|---|---|---|---|---|---|
| Tsai et al. [17] | 1994 | 1983–1993 | United States | Public | 15 | 10 | 5 |
| Samuels et al. [18] | 1996 | 1987–1995 | United States | Public | 14 | 7 | 7 |
| Blanche et al. [19] | 1997 | 1986–1995 | United States | Public | 30 | 18 | 12 |
| Miller et al. [20] | 1999 | 1987–1996 | United States | Public | 11 | 7 | 4 |
| Bacchetta et al. [21] | 2003 | 1993–2002 | United States | Public | 42 | 20 | 22 |
| Bridges et al. [22] | 2003 | 1997–2000 | United States and Canada | Public | 1092 | 551 | 541 |
| Edwards et al. [23] | 2003 | 1986–2000 | United Kingdom | Public | 35 | 17 | 18 |
| Levy Praschker et al. [24] | 2006 | 1990–2002 | France | Public | 30 | 11 | 18 |
| Roberts et al. [25] | 2006 | 2000–2006 | United States | Public | 9 | 6 | 3 |
| Hovanesyan et al. [26] | 2007 | 1996–2006 | United States | Public | 22 | 7 | 14 |
| Guilfoyle et al. [27] | 2008 | 1998–2007 | United Kingdom | Public | 23 | 13 | 10 |
| Ullery et al. [28] | 2008 | 1995–2004 | United States | Public/Private | 49 | 25 | 24 |
| Hajabed et al. [29] | 2010 | 2007–2009 | Saudi Arabia | – | 12 | 10 | 2 |
| Speziale et al. [30] | 2010 | 1998–2008 | Italy | Private | 127 | 62 | 65 |
| Easo et al. [31] | 2013 | 2000–2007 | Germany | Public | 17 | 6 | 11 |
| Assmann et al. [32] | 2013 | 1995–2011 | Germany | Public | 49 | 32 | 17 |
| Caceres et al. [33] | 2013 | 1983–2011 | United States | Public | 154 | 91 | 63 |
| Davis et al. [34] | 2014 | 2002–2012 | United States | Public | 108 | 61 | 47 |
| Murashita et al. [35] | 2014 | 1993–2013 | United States | Private | 33 | 12 | 21 |
| Mack et al. [36] | 2015 | 2007–2013 | United States | Public/Private | 20 | 8 | 12 |
| George et al. [37] | 2016 | 2001–2012 | United States | Public | 119 | 67 | 52 |
| Ohnuma et al. [38] | 2016 | 2011–2013 | Japan | Public | 34 | – | – |
| Tiwari et al. [39] | 2016 | 2004–2014 | United States | Public | 12 | 4 | 8 |
| Zack et al. [40] | 2017 | 2004–2013 | United States | Private | 1152 | 606 | 546 |
| Elgendy et al. [41] | 2019 | 2012–2014 | United States | Public | 840 | 479 | 361 |
| Elsisy et al. [42] | 2021 | 1993–2019 | United States | Private | 134 | 66 | 68 |
| Khalid et al. [43] | 2022 | 2012–2019 | United States | – | 2205 | – | – |
| Weinberg et al. [44] | 2022 | 2012–2019 | Australia | Public/Private | 18 | 14 | 4 |

## Risk of bias within the studies

The 28 publications selected for review were all retrospective cohort studies. According to the Scottish Intercollegiate Guideline Network cohort study checklist, no studies were classified as "high-quality" evidence. However, all were deemed to be of "acceptable quality," because they were based on clear and focused research questions.

## Study population

A total of 6411 patients were included in the review, all of whom were ≥90 years of age and underwent CABG, valvular surgery, aortic dissection repair, or a combination thereof (distribution of surgery types in Fig 2). Of the 28 publications selected for review, 19 included patients who underwent a combination of surgical interventions: six focused solely on surgical aortic valve repair, one focused exclusively on CABG, and one focused on repair of type A aortic dissection. Detailed sociodemographic data beyond these descriptors was not provided.

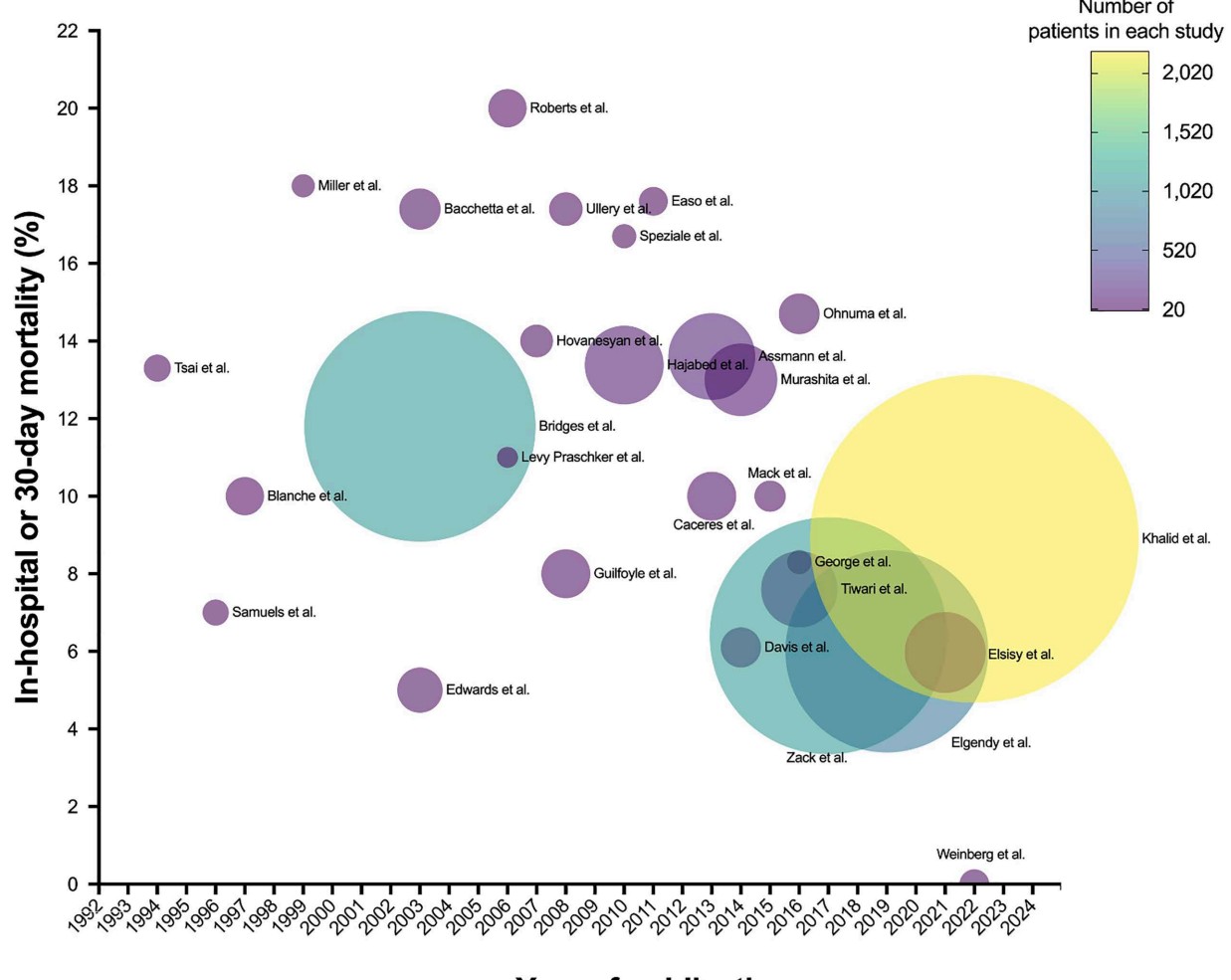

**Fig 2. Bubble plot illustrating in-hospital or 30-day mortality rates (%).** Each bubble represents a study, and bubble size corresponds to the number of patients included in the study (as indicated by the color gradient scale).

### Cardiopulmonary bypass and aortic clamp times

Whereas 15 publications reported the cardiopulmonary bypass (CPB) time, 13 reported the aortic cross-clamp (ACC) time. The median CPB time was 102 min (range, 60–152 min; IQR, 90–126), whereas the median ACC time was 60 min (range, 45–95 min; IQR 51.7–83.6). The CPB and ACC times in these studies are listed in S3 Table.

### Risk stratification scores

Among the 21 studies (77.8%) reporting some form of a risk stratification score, 19 studies (70.4% of total) used the New York Heart Association functional classification, four studies (14.8% of total) used EuroScores, three studies (11.1% of total) used the American Society of Thoracic Surgeons (STS) score, and one study (3.7% of total) used the Charlson Comorbidity Index.

### Primary outcomes

**Mortality.** All 28 studies evaluated in-hospital or 30-day mortality rates. The median 30-day mortality rate was 10.5% (IQR 7.2–14.6). Nineteen studies also reported longer-term mortality rates. The in-hospital or 30-day mortality rates,

together with the corresponding number of patients included in each study are presented graphically in Fig 2. The inpatient, 30-day, 1-year, 2-year, and overall mortality rates are shown in Fig 3. Mortality stratified by each study, together with the types of cardiac surgical procedures performed, is presented in Fig 4.

**Incidence of complications.** Of the 28 (71.4%) studies, 20 reported postoperative complication rates; 3545 patients were included in this group, and the median overall complication rate was 71.4%. The reported complications were grouped into specific categories, including respiratory, cardiac, neurological, gastrointestinal, renal, infectious, arrhythmogenic, and hemorrhagic complications, as well as those requiring insertion of a permanent cardiac pacemaker. Respiratory, cardiac, renal, neurologic (i.e., confusion and disorientation), gastrointestinal, and/or infectious complications were reported in 19%, 20%, 14%, 18%, 5%, and 9% of these patients, respectively. Hemorrhagic and arrhythmogenic complications were reported in 9% and 45% of these patients, respectively, and 10% required pacemaker insertion. The overall complication rates by complication type are presented in S4 Table.

## Secondary outcomes

**Length of hospital and intensive care unit stay.** Fifteen publications reported the length of hospital stay, whose median was 12.5 days (IQR 10.3–18.0). Nine publications reported the length of ICU stay, whose median was 7.1 days (IQR 3.3–11.5) (S5 Table). No studies reported unplanned ICU readmissions.

**Patient-reported experience and outcomes measures.** None of the included studies reported PROMs or PREMs; therefore, this study objective was unmet.

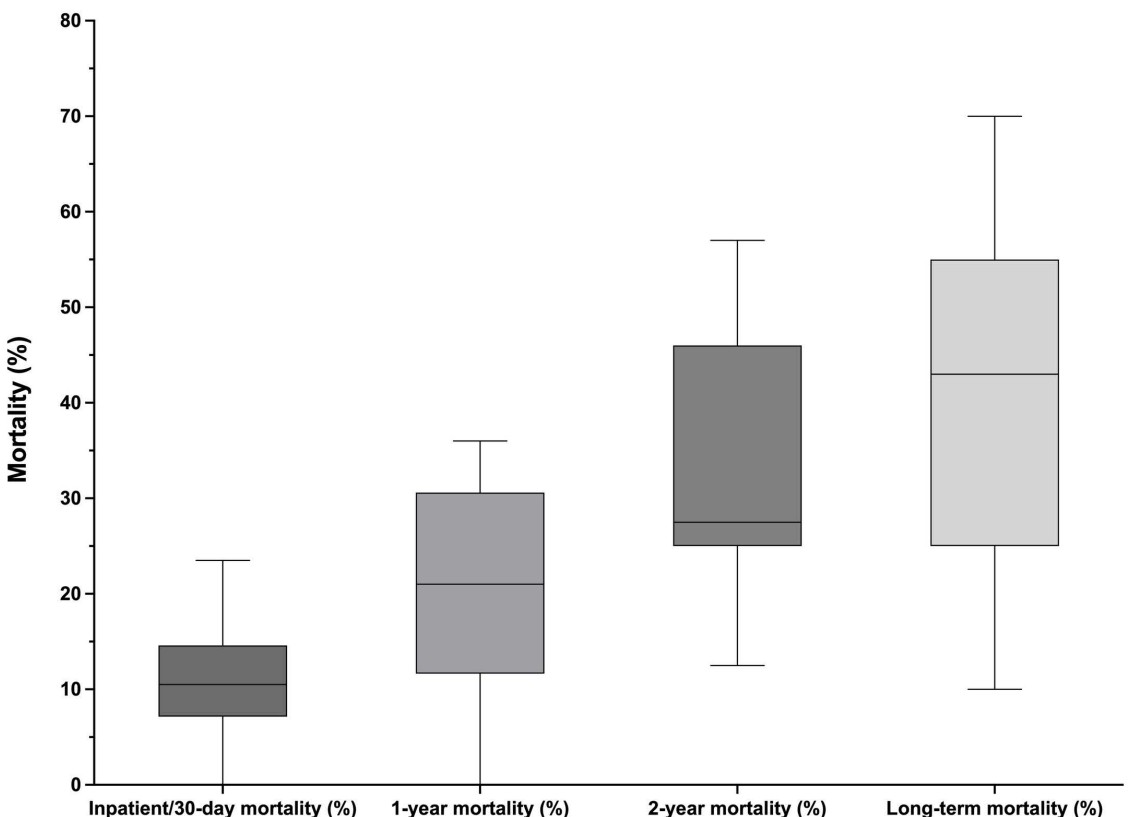

**Fig 3. Box and whisker plots showing inpatient, 30-day, 1-year, 2-year, and overall mortality rates.** The box shows the median and 25th and 75th quartiles. The whiskers represent the minimum and maximum values.

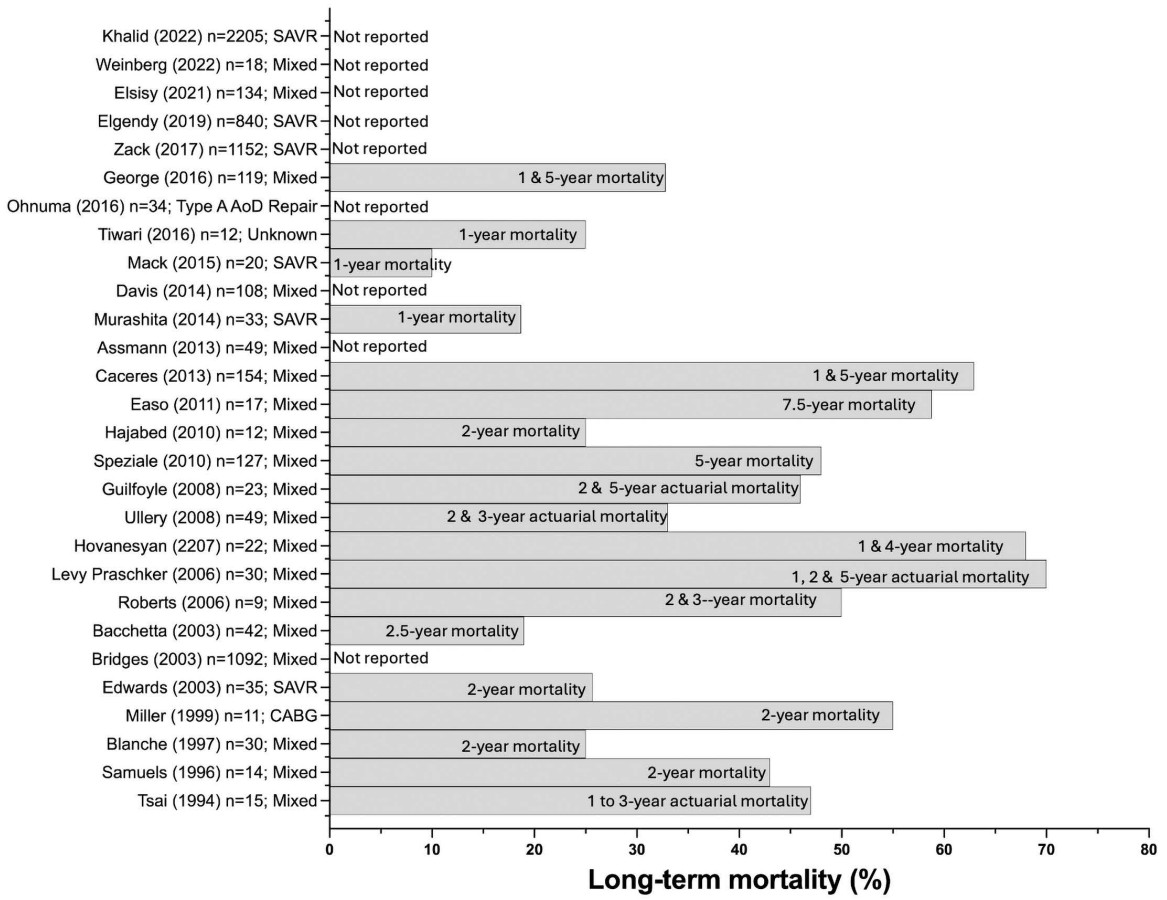

**Fig 4. Mortality stratified by each study, together with the types of cardiac surgical procedures performed.** Abbreviations: SAVR, surgical aortic valve replacement; CABG, coronary artery bypass; type AoD, type A aortic dissection repair; mixed, aortic, mitral, or tricuspid valve repair or replacement (or a combination thereof).

## Discussion

As global life expectancy increases, and the nonagenarian population is predicted to exceed 30 million people by 2030 [2], the prevalence of symptomatic coronary and valvular heart diseases continues to increase [5,6]. Although percutaneous and transcatheter interventions are frequently evaluated in this age group, a notable proportion of these patients are not anatomically or technically suited for these procedures. In such cases, traditional cardiac surgery has become the preferred surgical option, regardless of patients' advanced age. Consequently, the number of nonagenarians considering and undergoing surgical interventions will continue to increase. This scoping review provided a detailed synopsis of the current understanding of the perioperative course, complications, and the effects of these procedures on patients' quality of life. An accurate understanding of the outcomes experienced by nonagenarians who have undergone cardiac surgery will be crucial for preoperative decision-making and appropriate informed consent [45].

Obtaining informed consent poses a major ethical challenge, particularly in the context of patients with cognitive decline or frailty, because these conditions can impair decisional capacity. Consequently, a nuanced approach is required to carefully balance respect for patient autonomy with the principle of beneficence. Ethical decision-making in this context also requires addressing broader considerations, such as avoiding ageism, ensuring equitable resource allocation, and upholding principles of justice. Central to this process is the use of patient-centered strategies grounded in comprehensive

geriatric assessments, which provide a holistic understanding of patients' medical, functional, and psychosocial needs to guide ethical and equitable care. Effective communication and shared decision-making processes are essential for navigating these complexities, and ensuring that care aligns with patients' values and preferences while optimizing outcomes. [45,46].

## Specific outcomes

**Mortality.** Whereas all studies reported in-hospital or 30-day mortality, only 66% included detailed reports of long-term mortality. Overall, the median in-hospital and 30-day mortality rates were below 11%, in agreement with the previously reported mortality rates of 7%–18% [31]. Interestingly, these mortality rates are somewhat lower than the reported rates of approximately 33% for elective cases and as high as 38% for emergency cases among nonagenarians undergoing high-risk non-cardiac surgery [10]. Several factors beyond the statistical methods are likely to have contributed to this counterintuitive finding. First, the stringent patient selection for cardiac surgery in nonagenarians creates a highly selected cohort: patients must demonstrate adequate physiological reserves, cognitive function, and life expectancy to justify the procedural risk. In contrast, non-cardiac surgery populations might include emergent presentations with less rigorous preoperative optimization. Second, the availability of transcatheter alternatives might selectively direct higher-risk nonagenarians away from open surgery. Frail patients might be triaged toward transcatheter aortic valve replacement or medical management, thus further contributing to the lower observed mortality. Third, the reported increase in mortality of nonagenarians undergoing non-cardiac surgery represents a general nonagenarian population rather than a highly selected cardiac-surgery cohort. Fourth, the data reflect predominantly high-income countries with advanced perioperative care systems, specialized cardiac surgery centers, and comprehensive postoperative support.

Although older patients are widely understood to be likely to have poorer surgical outcomes and experience greater morbidity than their younger counterparts, our findings suggest that nonagenarians undergoing cardiac surgery might have lower in-hospital and 30-day mortality rates than nonagenarians undergoing non-cardiac surgery [10]. The apparent reversal might reflect the intersection of careful patient selection, advanced healthcare infrastructure, and the inherent bias in studying only patients deemed suitable for major cardiac surgery. Collectively, the included studies reported a median long-term mortality rate of 43%, and a subset of studies reported an average 5-year mortality rate of 53%. As noted above, long-term mortality data were reported at time points ranging from 1 to 7.5 years. The lack of a uniform definition or method for assessing long-term mortality among nonagenarians undergoing cardiac surgery might create difficulties for anesthetists, physicians, surgeons, and patients seeking to weigh the potential benefits of surgery against the optimal medical management of their cardiovascular disease.

**Complications.** Of the 28 publications, 20 reported complications experienced by nonagenarians undergoing cardiac surgery, and revealed a median complication rate of 71%, a value exceeding the 60% reported for major non-cardiac surgery [10]. The most frequently reported complications were arrhythmogenic complications, with an incidence of 45%, similarly to previously published findings [34]. Collectively, these findings suggest that atrial fibrillation occurs more frequently in nonagenarians than in younger patients undergoing cardiac surgery [25]. Although previous studies have highlighted the efficacy of prophylactic antiarrhythmic medications in decreasing postoperative atrial tachycardia in younger cardiac surgical patients, the generalizability of these findings to the nonagenarian population remains unclear [47]. The perioperative administration of these medications was not addressed in any studies included in this review. Moreover, a study published in 2023 by Gaudino et al. [48], including more than 1.2 million patients, has reported a coronary artery bypass grafting complication rate of only 15% among patients 59–73 years of age. Similarly, Bis et al. [49], in a study on postoperative conduction disturbances that included 15,000 patients within a similar age range, have found that permanent pacemaker implantation was required in only 1%–5% of cases. Respiratory, renal, and neurological (e.g., delirium) complications had reported median rates of 19%, 14%, and 18%, respectively. These complications frequently develop in vulnerable surgical populations [50].

The observed elevated complication rates have major implications for surgical decision-making in this cohort. Although advanced age alone should not preclude surgical intervention, the heightened risk of complications and mortality warrants careful consideration. The extant literature has reported postoperative complication rates reaching 60% in nonagenarians undergoing major surgery, and a 1-year mortality rate of approximately 20% after cardiac procedures. These substantial risks necessitate comprehensive discussions with patients and their families to facilitate informed decision-making. Surgeons must carefully balance the potential benefits of surgery against the associated risks, while considering patient preferences and anticipated postoperative quality of life. In assessing surgical candidacy, factors such as frailty, comorbidities, and functional status should be evaluated in conjunction with age. Moreover, the urgency of the procedure is a critical consideration, because elective surgeries generally yield more favorable outcomes than urgent or emergent surgeries. Ultimately, the decision to proceed with cardiac surgery in nonagenarians should be individualized, taking into account not only survival prospects but also the potential for improved functional status and quality of life.

**Length of hospital and ICU stay.** ICU admission can have adverse physical, cognitive, and psychological effects on patients, which are frequently referred to as post-intensive care syndrome. This syndrome can lead to significant long-term adverse sequelae for patients undergoing surgery, as well as their families and caregivers [50]. Although the length of ICU stay is directly associated with diminished survival rates and quality of life [51], only five (18%) studies addressed both the length of hospital stay and ICU admission. Of the original 28 publications included in this study, nine (33%) reported the length of ICU admission, and 15 (54%) reported the length of hospital stay. Given the inconsistencies among these reports, the results of cross-study comparisons are unlikely to be highly reliable. The reported median ICU and hospital stays of 7.0 and 12.5 days, respectively, were longer than those previously reported for octogenarians [52]. Better understanding of the factors contributing to these prolonged admissions is needed to improve informed decision-making and optimize the selection of nonagenarians most likely to benefit from surgery.

**Quality of life outcome measures and frailty.** Whereas all studies included in this review reported in-hospital or 30-day mortality rates, nearly one-third did not report long-term mortality rates. Less than one-third of the studies included quality-of-life metrics but used inconsistent reporting standards [22]. Furthermore, baseline function and comorbidities were reported in only 18 of the original 28 studies.

Our results revealed a notable scarcity of literature examining the effects of frailty on cardiac surgery outcomes in nonagenarians. Indeed, nonagenarians in relatively poor health might potentially have been steered away from these procedures by physicians, family members, or themselves. Overall, we identified no consistent use of specific frailty scores, and analyses of the effects of frailty on patient outcomes after cardiac surgery were limited. Similarly, long-term age-related health status following major noncardiac surgery remains largely unaddressed. In one small study including 159 octogenarians and nonagenarians who underwent non-cardiac surgery, only 50% of the patients survived for more than 3 years [53]. Whereas our review of recent evidence suggested that good surgical outcomes can be achieved with careful patient selection, and that age alone should not be a barrier to surgery, a previous study has reported that 25% of patients older than 85 years undergoing noncardiac surgery experienced a moderate, severe, or total limitation in functional capacity [54]. Our scoping review revealed a substantial gap in the literature: very few reports have focused specifically on functional outcomes in older patients who have undergone cardiac surgery.

## Implications and knowledge gaps

Our review summarized the body of evidence and crucial outcome indicators and shed light on the unique challenges faced by nonagenarians after cardiac surgery. Collectively, our findings underscored that age alone should not be a disqualifying factor for surgery. With careful selection, this patient group experiences a relatively low perioperative 30-day mortality rate, thus highlighting the importance of considering postoperative complications and prolonged ICU stays. Although we extracted elective versus emergency classification, when provided, the reporting was inconsistent; therefore,

improved standardization is warranted in future research. Whereas the 1-year mortality rate was approximately 20%, the lack of PROM and PREM data suggested that post-surgical quality of life remains largely undetermined.

This review emphasizes the need to standardize complication grading and reporting in cardiac surgery for nonagenarians (Table 3). The findings also highlight the need for prospective trials and reviews of international registry data to identify risk factors associated with adverse outcomes in nonagenarians, explore potential measures to decrease the risk of severe complications, and improve the quality of life after surgery.

### Strengths and limitations

This review has several notable strengths. First, it provided a comprehensive analysis of mortality and complication rates in nonagenarians undergoing cardiac surgery, thereby offering valuable insights into the outcomes associated with these high-risk procedures. Additionally, the findings underscored critical gaps in the current literature and suggested several directions for future research that might enhance understanding of the postoperative trajectory in nonagenarian patients. Such advancements might potentially inform clinical decision-making and optimize care strategies for this vulnerable population. The findings might also be used to facilitate informed preoperative discussions and decision-making by physicians, surgeons, anesthetists, and, most importantly, patients and their families.

This review has several limitations. First, patient-level data were not reported in many of the included studies. Therefore, analyses could be performed only on reported population data, which provided no insights into specific pre-existing comorbidities, frailty, or the urgency of surgical intervention. Unfortunately, most included studies did not stratify outcomes by surgical urgency, thus constraining our ability to evaluate the differential effects of elective vs. emergent procedures. Furthermore, although we collected data reported for 6411 nonagenarians included in 28 publications, two studies [40,43]

**Table 3. Recommended minimum standards for reporting on nonagenarians undergoing cardiac surgery.**

| Preoperative | Intraoperative | Postoperative |
|---|---|---|
| • Anthropometrics: weight, height, body mass index<br>• Residential status: home, residential care facility, level of dependence<br>• Rurality and socio-economic status<br>• Frailty score [55]<br>• Comorbidities<br>• EuroSCORE II<br>• Assessment for cognitive impairment and dementia [56]<br>• Echocardiographic findings: ventricular function, pulmonary artery pressures<br>• Time from hospital admission to surgery<br>• Blood tests: hemoglobin, creatinine, albumin, fibrinogen, coagulation, ferritin, troponin (mmol/L), B-type natriuretic peptide (pg/mL) | • Surgery category: elective, urgent, emergent<br>• Type of surgery<br>• Cardiopulmonary bypass time<br>• Cross-clamp time<br>• Use of blood or blood products<br>• Need for vasoactive medication and inotropes<br>• Need for intraoperative mechanical support | • Duration of mechanical ventilation<br>• Duration of epicardial pacing time<br>• Use of blood or blood products<br>• Return to operating theater<br>• Standardized definitions of postoperative complications [57,58] graded according to Clavien-Dindo [59] classification<br> - Acute kidney injury<br> - New atrial fibrillation and/or atrial flutter<br> - Delirium<br> - Pulmonary complications [60]<br> - Stroke<br> - Surgical site infection<br> - Return to operating theatre and reason<br> - Heart block and/or bradycardia requiring pacemaker<br> - In-hospital mortality<br> - Unexpected readmission to ICU<br> - Hemoglobin and creatinine values on hospital discharge<br>• Discharge destination: home or rehabilitation facility<br>• Quality of recovery scores [60,61]<br>• Postoperative morbidity survey [62–64]<br>• Stroke within 30 days of surgery<br>• Days alive and at home, as many as 30, 90 or 180 days after surgery [65–68]<br>• Unplanned hospital readmissions<br>• Patient-reported outcome measures (PROMs) [68–69]<br>• Patient-reported experience measure (PREMs) [70]<br>• Health economic measures |

performed in the past 2–6 years accounted for more than half of those patients, and included those most likely to benefit from the most recent developments in this field. We acknowledge that the patients included in our review from The Society of Thoracic Surgeons National Database [23] might overlap with patients in the nine other studies included in this review. This potential crossover is an important consideration in interpreting our findings. However, after exclusion of these studies from the analysis, our results were unchanged. This finding further highlights the need for future research to more precisely quantify the extent of patient overlap in multi-center cardiac surgery studies. Third, because of the apparent heterogeneity in comorbidity and functional status, we were unable to generate strong conclusions regarding preoperative status and its potential effects on postoperative outcomes and quality of life. Because no sociodemographic data were available, the generalizability of our findings across sexes, ethnicities, regions, and countries of origin remains unclear.

Finally, we acknowledge that a substantial proportion of the articles included in our analysis were older than 10 years and consequently might not fully reflect current medical practices, particularly given the increasing prominence of endovascular procedures in treating nonagenarian patients across various cardiovascular domains, including structural valve pathology. However, the included studies provide historical context and insight into the evolution of cardiac surgery in nonagenarians; highlight advancements and shifts in treatment paradigms over time; and offer a broader perspective on long-term outcomes and complications, which are crucial for understanding the durability and effectiveness of interventions. Finally, our findings cannot be generalized to lower-resource settings or lower- and middle-income countries, where risk profiles and outcomes may differ significantly.

## Ethical tensions, individual benefits, and resource allocation

Providing open cardiac surgery to nonagenarians presents complex health economic and ethical challenges, particularly in resource-constrained settings and lower- and middle-income countries, where healthcare systems often operate with limited capacity and funding. The cost of cardiac surgery is substantial, and prolonged ICU stays, increased complication rates, and extended hospital admissions are often involved, thereby contributing to higher healthcare expenditures than required for younger cohorts. Moreover, in lower- and middle-income countries, where access to expensive interventions must be weighed against broader public health needs, questions arise regarding whether prioritizing high-cost procedures in the oldest populations constitutes an equitable or sustainable use of limited resources. Consequently, tension exists between the ethical principle of individual beneficence, providing the best possible care to a single patient, and the utilitarian imperative to maximize health outcomes at the population level. Ivashkov and Van Norman have argued that surgical decision-making in older adults must balance respect for autonomy with realistic expectations of benefit and resource stewardship, particularly when quality of life might be diminished by frailty, comorbidities, or cognitive impairment [71]. Similarly, Altawalbeh et al. have highlighted how age-related ethical frameworks must incorporate considerations of distributive justice, given that the opportunity cost of providing intensive treatment to nonagenarians might result in denying care to younger or more functionally independent patients [46]. Because surgical outcomes in the oldest populations remain highly variable, and data on long-term functional recovery remain sparse, healthcare policymakers must increasingly confront the moral dilemma of whether aggressive interventions are justifiable when life expectancy and quality-adjusted life years might be limited. Accordingly, transparent, culturally sensitive, and evidence-informed triage frameworks are essential to support ethical and economically sound decision-making in cardiac surgery for the oldest populations. As demographics shift globally, policies must reconcile economic realism with ethical imperatives to avoid discriminatory rationing, and must uphold the "fair innings" principle aimed at equitable health opportunities across lifespans [72,73].

## Future directions

This scoping review highlights several important avenues for future research. Among these, establishing a standard time point for assessing long-term mortality, and applying and validating objective pre- and postoperative functional status and

quality-of-life metrics, might be beneficial [74,75]. Such initiatives stand to greatly improve understanding of the short- and long-term effects of cardiac surgery in nonagenarian populations beyond immediate survival and complication rates.

Creating and implementing uniform preoperative frailty assessments, and correlating them with surgical outcomes, will also be crucial. Assessments should include preoperative residential status, hospital discharge destination, and independence in activities of daily living 12 months post-cardiac surgery. In addition, the development of standardized PROMs and PREMs tailored to nonagenarians is imperative. These tools provide healthcare providers with essential data to evaluate the cost-effectiveness of surgical interventions in terms of quality-adjusted life years, thereby offering a common value currency for comparing different disease states [76].

Future research should prioritize investigating the specific effects of individual cardiac procedures, particularly in nonagenarian populations, to provide more nuanced insights beyond the aggregated outcomes of all open cardiac interventions. Such studies would offer valuable evidence to support informed decision-making tailored to this unique demographic. Additionally, the effects of cardiac surgery on functional status and quality of life are critical areas for further exploration, because these outcomes are central to patient-centered care. Leveraging data from large national registry databases, such as SWEDEHEART [77], presents opportunities to expand upon existing findings, and generate robust evidence to inform clinical practice and policy.

Finally, the findings presented in this review might assist cardiac surgeons, anesthetists, and perioperative clinicians in assessing both the quantifiable risks and functional unknowns of cardiac surgery, and further provide patients and their families with critical information to guide discussions focused on the benefits and risks of proceeding with cardiac surgery or seeking alternative management strategies.

## Conclusions

This scoping review highlights that, although age alone should not be a barrier to cardiac surgery for nonagenarians, this surgery is associated with significantly elevated morbidity and a relatively high mortality rate. Notably, among patients in this age range, one in ten die, and two-thirds of patients succumb to complications within 30 days post-surgery. Most of these patients experience extended intensive care stays. Despite these challenges, most of these patients are ultimately discharged from the hospital. A critical gap in the literature is the scarcity of data on quality-of-life measures and patient-reported outcomes, which are essential for informed clinical discussions and consent. Incorporating these factors into decision-making processes will be crucial to ensure that care aligns with patients' values and preferences, particularly in this vulnerable population. Future research should prioritize the collection and analysis of these outcomes to enhance the quality and relevance of care for nonagenarians undergoing cardiac surgery.

Our review highlights major gaps in the collective knowledge base and specific areas requiring further research, including standardized methods for reporting complications, patient-centered outcomes, functional recovery metrics, and long-term mortality data. Addressing these gaps would enhance understanding of patient selection criteria, as well as the perioperative and postoperative experiences of nonagenarians undergoing cardiac surgery.

## Supporting information

**S1 Table. Preferred Reporting Items for Systematic reviews and Meta-Analyses extension for Scoping Reviews (PRISMA-ScR) Checklist.**
(DOCX)

**S2 Table. Full search strategies for all electronic databases.**
(DOCX)

**S3 Table. Cardiopulmonary bypass and aortic cross-clamp: times reported by the publications included in the review.**
(DOCX)

**S4 Table. Overall complication rates listed by type.**
(DOCX)

**S5 Table. Length of hospital and intensive care stay.**
(DOCX)

## Guarantor

Prof. Laurence Weinberg is the guarantor.

## Author contributions

**Conceptualization:** Laurence Weinberg, Jarryd Ludski, Bradly Carp, Je Min Suh, Anoop N Koshy, Cilla Haywood, Benjamin Churilov, Dong-Kyu Lee, Michael Yii.

**Data curation:** Laurence Weinberg, Jarryd Ludski, Bradly Carp, Dong-Kyu Lee.

**Formal analysis:** Laurence Weinberg, Jarryd Ludski, Bradly Carp, Dong-Kyu Lee.

**Investigation:** Laurence Weinberg, Jarryd Ludski, Je Min Suh, Cilla Haywood.

**Methodology:** Laurence Weinberg, Jarryd Ludski, Bradly Carp, Je Min Suh, Anoop N Koshy, Cilla Haywood, Benjamin Churilov, Dong-Kyu Lee.

**Supervision:** Laurence Weinberg.

**Writing – original draft:** Laurence Weinberg, Jarryd Ludski, Bradly Carp, Je Min Suh, Anoop N Koshy, Cilla Haywood, Benjamin Churilov, Dong-Kyu Lee, Michael Yii.

**Writing – review & editing:** Laurence Weinberg, Jarryd Ludski, Bradly Carp, Je Min Suh, Anoop N Koshy, Cilla Haywood, Benjamin Churilov, Dong-Kyu Lee, Michael Yii.

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
