## [Decision Letter · Decision Letter 0]

13 Nov 2024

Dear Dr. Weinberg,

**ACADEMIC EDITOR:**

While the manuscript demonstrates significant potential, it requires major revisions and further refinement to meet the standards necessary for publication.

The subject matter is undoubtedly of interest; however, the reviewers have raised critical concerns that must be thoroughly addressed by the authors.

We look forward to receiving your revised manuscript.

Kind regards,

Marcelo Arruda Nakazone, M.D., Ph.D.

Academic Editor

PLOS ONE

Journal Requirements:

https://bmjopen.bmj.com/content/13/7/e072293.full

In your revision ensure you cite all your sources (including your own works), and quote or rephrase any duplicated text outside the methods section. Further consideration is dependent on these concerns being addressed.

3. We note that there is identifying data in the Supporting Information file <Supplementary Table 2 .docx>. Due to the inclusion of these potentially identifying data, we have removed this file from your file inventory. Prior to sharing human research participant data, authors should consult with an ethics committee to ensure data are shared in accordance with participant consent and all applicable local laws.

-Location data

Additional guidance on preparing raw data for publication can be found in our Data Policy (https://journals.plos.org/plosone/s/data-availability#loc-human-research-participant-data-and-other-sensitive-data ) and in the following article: http://www.bmj.com/content/340/bmj.c181.long .

Please remove or anonymize all personal information (Name/Initials), ensure that the data shared are in accordance with participant consent, and re-upload a fully anonymized data set. Please note that spreadsheet columns with personal information must be removed and not hidden as all hidden columns will appear in the published file.

Reviewers' comments:

Reviewer's Responses to Questions

**Comments to the Author**

1. Is the manuscript technically sound, and do the data support the conclusions?

Reviewer #1: Yes

Reviewer #2: No

Reviewer #3: Partly

2. Has the statistical analysis been performed appropriately and rigorously?

Reviewer #1: Yes

Reviewer #2: N/A

Reviewer #3: No

3. Have the authors made all data underlying the findings in their manuscript fully available?

Reviewer #1: Yes

Reviewer #2: Yes

Reviewer #3: Yes

4. Is the manuscript presented in an intelligible fashion and written in standard English?

Reviewer #1: Yes

Reviewer #2: Yes

Reviewer #3: No

Reviewer #1: This systematic review on outcomes and complications in nonagenarians undergoing cardiac surgery addresses a relevant and timely topic. It provides important insights into mortality rates, complications, and hospital stays in this age group. However, there are several areas where the manuscript could be improved.

Strengths:

The topic is highly pertinent given the aging population and increasing number of elderly surgical candidates. The literature search is thorough, covering multiple countries and procedures. PRISMA guidelines were followed.

Minor Concerns:

The introduction could be streamlined to focus more on the specific gaps in the literature that this review addresses.

The rationale for excluding non-English articles and certain cardiac procedures could be more clearly explained, as this may limit the generalizability of findings. The statistical methods are not adequately described. More detail on data synthesis and analysis, particularly handling of heterogeneity (if any) among studies, is needed. The statistical methods used for data synthesis may be described in more detail. While the manuscript mentions the use of descriptive statistics, there is a lack of detail on how these were calculated and whether any meta-analytic techniques were applied. The authors could consider providing a more robust analysis or explaining why certain methods (e.g. meta-analysis) were not feasible.

The discussion should more critically evaluate the findings, particularly the high complication rates, and offer a more balanced view on surgical risks and benefits. For example, the implications of the high complication rates could be explored in greater depth, particularly concerning how these might influence surgical decision-making in nonagenarians. The manuscript suggests that age should not be a disqualifying factor for surgery, which is an important point. However, this statement would benefit from a discussion of the balance between risks and benefits, considering patient preferences and quality of life post-surgery.

The manuscript identifies a lack of patient-reported outcomes but could suggest how future research might address this gap.

There are a few typographical errors and grammatical issues throughout the manuscript. A thorough proofreading would be beneficial. For example, the sentence "From the initial 1015 articles searched, we included twenty-eight studies from eight countries were included" is redundant and should be revised for clarity.

Ethical Considerations:

A discussion of the ethical implications of surgery in nonagenarians, particularly around informed consent, would be valuable.

Reviewer #2: 1. The authors have to make a choice between a systematic and a scoping review that they seem to assimilate (even in title) but which are two different type of studies with different aim and methodology.

Munn, Z., Peters, M.D.J., Stern, C. et al. Systematic review or scoping review? Guidance for authors when choosing between a systematic or scoping review approach. BMC Med Res Methodol 18, 143 (2018). https://doi.org/10.1186/s12874-018-0611-x.

2. One strength of the study is the extended review of literature and the rigorous methodology deployed to perform the analysis since out of 1015 articles searches (with no restriction on study design or date of publication), the authors included 28 studies from 8 countries (most of them issued from US centers) and totalizing a sample size of 6411 patients.

However, the majority of those papers are older than 10 years and certainly do not reflect medical practice in the current era where endovascular procedures has become the main treatment carrier in nonagenarian patients in all domains of cardiovascular diseases including structural valve pathology. Those papers totalize 17% of the overall population but it is not specified how did the authors ensure that patients included in the more recent studies of Khalid, Zack and Elgendy, did not already appear in previous published studies from US. Therefore, in the current review, the probability for one patient to have been reported at least twice is significant.

3. Since a scoping review focuses on the importance of one specified subject in the literature, it is expected that the outcome would relate on one bibliometric index such as the annual incidence of papers related to the subject under study. Instead the authors have used traditional clinical outcomes such as in-hospital mortality, morbidity and length of stay. It even appears that they may have discarded some relevant papers which did not included those clinical outcomes. This is deeply confounding. If they authors aim to perform a systematic review or a metanalysis of the clinical outcomes of cardiac surgery in nonagenarian patients, then the title must be changed as well as the paper has to be deeply revised accordingly.

4. The discussion and interpretation reflect more the author’s belief than the result that they present. Based on those results I would rather assume that the relatively seldom number of papers found in the literature concerning cardiac surgery in nonagenarian patients is a marker of the reluctance to propose chest opening and CPB to frail patients with limited life expectancy. Most of them would probably deny it anyway unless they have the feeling of an imminent death or, most likely, even though. If the well-looking conclusion (not to deny cardiac surgery because of patient’s age only) seems quite reasonable, it is hardly supported by the presented results for a 10% mortality rate for elective procedures is not deemed acceptable in today standards of cardiac surgery.

Reviewer #3: The authors present a scoping review of the literature to look at the outcomes of cardiac surgery in nonagenarians. They have published their protocol and registered the review. The standard of English is generally good although there are scattered typos and grammatical errors that require a thorough check through the paper (e.g. line 138, redundant "The"s). The following comments are designed to help the authors address some conflicts between protocol and paper.

1. There is differing definitions of the Population. In some sections, it is >90y, in others 90-99. Clarify which - and in cases where papers presented mixed data, how this was resolved

2. The paper suggests it will include grey literature, but largely excludes by design the large national registries that might describe this information (e.g., although outside the search dates, the SWEDEHEART registry of nonagenarians published recently)

3. Line 148 - please clarify, were full papers translated?

4. Line 151 - please clarify, this sentence is ambiguous

5. Line 156 - centenarians included here

6. Line 159 - lots of procedures not in the search criteria

7. Information sources are not outlined as per PRISMA giuidelines - how were authors contacted etc

8. Line 193 - not clear how two reviewers can agree on four and five studies and end with nine studies? Surely agreement would mean including just four studies (and one that was not independently agreed on)

9. Line 239 - PPI is possible even in reviews

10. Ref 23 - this large study from the STS database likely crosses over some of the patients from the other studies. How did the authors account for this?

11. Line 274 - Figure 2 does not show operations, but this information would be useful and is not shown elsewhere

12. 278 - was additional detail provided in none of the 28 studies? This seems a surprise

13. 289 - was risk stratification also provided e.g. EuroSCORE or STS score?

14. Fig 2 - this figure is complex. Would a bubble plot help to show the same data in a more condensed form, with the size of the bubble indicating the size of the study, x being time and y being survival?

15. Fig 3 - this summary figure was much more helpful and easily to interpret

16. 303-313 - complication rates are very difficult to assess when presented in isolation; some of these are common in any age group, some may be likely in certain procedure groups. A more descriptive narrative of the findings would help the reader.

17. The median time for long term mortality follow up needs to be described as this is very subjective in this age group

18. 383 and 317 report different median stays

19. It was not clear what the outcome of the thematic analysis was. Ideally report this in a separate section.

20. Table 3 - is there any way of summarising how many of the studies included reported this minimum dataset?

21. The statistical method of weighting the different studies is not outlined. Please explain how studies contributed to the averages shown.

Overall, I found this study interesting but with several concerns about the precise methodology, as above. I am not an expert in scoping reviews, so have referred to the PRISMA guidance to assess this, but have the impression that this has been largely adhered to. Where there is discrepancy, I have sought to identify where improvements could be made. The relative paucity of experience with scoping reviews in the cardiac surgery literature is not a fault of the authors, but rather one of this reviewer. Nonetheless, the net effect is that the authors must tolerate a more didactic review.

**Do you want your identity to be public for this peer review?** For information about this choice, including consent withdrawal, please see our Privacy Policy

Reviewer #1: **Yes: ** Yousef Tanas

Reviewer #2: **Yes: ** Pr Thierry CAUS

Reviewer #3: No

---

## [Author Response · Author response to Decision Letter 1]

31 Mar 2025

Professor Marcelo Arruda Nakazone, M.D., Ph.D.

Academic Editor

PLoS ONE

Ref: Resubmission of PONE-D-24-26035

Title: Outcomes and complications in nonagenarians undergoing cardiac surgery: a scoping review

Dear Dr. Nakazone:

My coauthors and I would like to thank you and the expert reviewers for taking the time to provide a very constructive and thoughtful review of our manuscript.

As requested, we have included the following items with our resubmission:

1. A point-by-point response letter (to follow) that addresses each issue raised by the Academic Editor and the Reviewers. I have uploaded this letter as a separate file labelled “Response to Reviewers.”

2. A marked-up copy of the manuscript that highlights changes made to the original version. This has been uploaded as a separate file labelled “Revised Manuscript with Tracked Changes.”

3. An unmarked version of our revised paper without tracked changes. This has also been uploaded as a separate file labelled “Manuscript.”

Journal Requirements

Question 1: Please ensure that your manuscript meets PLOS ONE's style requirements, including those for file naming. The PLOS ONE style templates can be found at

Authors’ response to Q1: Thank you for this important comment. The text has been edited and formatted and is now consistent with the style guidelines provided above.

Question 2. We noticed you have some minor occurrence of overlapping text with the following previous publication(s), which needs to be addressed: https://bmjopen.bmj.com/content/13/7/e072293.full. In your revision ensure you cite all your sources (including your own works), and quote or rephrase any duplicated text outside the methods section. Further consideration is dependent on these concerns being addressed.

Authors’ response to Q2: The manuscript noted at this website (https://bmjopen.bmj.com/content/13/7/e072293.full), entitled “Outcomes and complications of nonagenarians undergoing cardiac surgery: a scoping review protocol” (Ludski et al.) is our earlier publication of the protocol used to guide the study described here. We have now cited this in the text) and also included this as reference #15. As noted in the text, we published the protocol to ensure more effective dissemination, public accessibility, and transparency. Of necessity, we repeated some of the text from this publication (as noted above, minor) to provide clarity throughout.

We are pleased to inform you that we have carefully rephrased any duplicated text in our manuscript titled “https://bmjopen.bmj.com/content/13/7/e072293.full; “Outcomes and complications of nonagenarians undergoing cardiac surgery: a scoping review protocol” (Ludski et al.) to ensure originality and clarity. Additionally, The University of Melbourne has conducted a thorough review using iThenticate to verify that the content does not overlap with our prior publications. The similarity index for the manuscript is below 10%, with all flagged matches appropriately attributed to cited sources. We are happy to provide this information if requested. We are confident these revisions align with the journal’s ethical standards and formatting guidelines.

Should the editorial team or reviewers recommend further adjustments, we are fully committed to addressing them promptly.

Question 3. We note that there is identifying data in the Supporting Information file <Supplementary Table 2 .docx. Due to the inclusion of these potentially identifying data, we have removed this file from your file inventory.

Authors’ response to Q3: Thank you for notifying us of this oversight.

Reviewer ONE

We thank Reviewer One for their expert insights and time spent reviewing our manuscript. We have provided a detailed comment to each of the reviewer’s questions, as outlined below.

1. Reviewer 1, comment 1: This systematic review on outcomes and complications in nonagenarians undergoing cardiac surgery addresses a relevant and timely topic. It provides important insights into mortality rates, complications, and hospital stays in this age group. However, there are several areas where the manuscript could be improved.

Strengths: The topic is highly pertinent given the aging population and increasing number of elderly surgical candidates. The literature search is thorough, covering multiple countries and procedures. PRISMA guidelines were followed.

Authors’ response to R1 comment 1: Thank you for these positive comments.

Minor Concerns

2. Reviewer 1, comment 2: The introduction could be streamlined to focus more on the specific gaps in the literature that this review addresses.

Authors’ response to R1 comment 2: Done. The Introduction has been streamlined and now focuses primarily on the goals of the scoping review. Thank you for this constructive comment.

3. Reviewer 1, comment 3: The rationale for excluding non-English articles and certain cardiac procedures could be more clearly explained, as this may limit the generalizability of findings.

Authors’ response to R1 comment 3: We have expanded our rationale for excluding articles not written in English (note that we did include articles with English-language abstracts and had the full papers translated if they fulfilled the study criteria). Interestingly, a 2021 study published by Dobrescu and colleagues in the Journal of Clinical Epidemiology reported that restricting selections to English-language publications appeared to have little impact on the effect estimates and conclusions of systematic reviews. This information has been included together with a new reference [16].

Also, we thought that rather than explaining the cardiac procedures (which would add quite a bit of extra text to the manuscript), the reader might refer to the original studies (references [17-44]) for further details. However, we have expanded the list of cardiac procedures as outlined in the revised manuscript.

4. Reviewer 1, comment 4: The statistical methods are not adequately described. More detail on data synthesis and analysis, particularly handling of heterogeneity (if any) among studies, is needed. The statistical methods used for data synthesis may be described in more detail. While the manuscript mentions the use of descriptive statistics, there is a lack of detail on how these were calculated and whether any meta-analytic techniques were applied. The authors could consider providing a more robust analysis or explaining why certain methods (e.g. meta-analysis) were not feasible.

Authors’ response to R1 comment 4: Thank you for this constructive comment.

The primary aim of our scoping review was to map the available evidence rather than generate a pooled effect estimate. Differences in outcomes or patient characteristics across studies were discussed narratively.

There were several reasons why meta-analytic techniques could not be used in our scoping review. All our included studies typically ranged from very small case series to large observational studies. There were no randomised controlled trials included. Of the 28 studies we included, 20 (71.4%) had less than 50 patients included. There were 12 studies (42.9%) with less than 25 patients. This methodological diversity makes it inappropriate to combine effect sizes statistically. More importantly, many of the outcome measure were variable. Studies reported outcomes such as complications using different definitions. Timeframes, measurement tools also varied preventing direct statistical comparison. The included papers comprised various types of cardiac surgeries, from coronary bypass to complex major aortic interventions, making a single pooled estimate meaningless.

Our data synthesis and analysis were conducted systematically to ensure rigor and reliability. All continuous data were tested for normality using the Kolmogorov–Smirnov test and/or visual inspection of the Q‐Q plot. For data that followed a normal distribution, we reported means and standard deviations, while for non-normally distributed data, we utilized medians and interquartile ranges [IQRs], and ranges (minimum-to-maximum values) to summarize the findings. Differences in categorical variables were assessed using either the Chi-square test or Fisher’s exact test, depending on the sample size and distribution of the data.

In studies where heterogeneity was evident, a narrative synthesis approach was planned to enhance understanding of patient-reported outcomes. This approach involved identifying research gaps through a comparative analysis of study designs and participant characteristics. Additionally, where feasible, thematic analysis was undertaken to explore references to quality-of-life outcomes among nonagenarians. The thematic analysis process included familiarization with the data, generation of codes to capture key concepts related to outcomes, complications, and management strategies. This methodological flexibility allows for the nuanced exploration of the complexity present in the literature while maintaining adherence to a coherent analytical framework. This framework facilitates the identification of knowledge gaps and the recognition of emerging trends within the field.

This has been updated in the resubmission manuscript.

5. Reviewer 1, comment 5: The discussion should more critically evaluate the findings, particularly the high complication rates, and offer a more balanced view on surgical risks and benefits. For example, the implications of the high complication rates could be explored in greater depth, particularly concerning how these might influence surgical decision-making in nonagenarians. The manuscript suggests that age should not be a disqualifying factor for surgery, which is an important point. However, this statement would benefit from a discussion of the balance between risks and benefits, considering patient preferences and quality of life post-surgery.

Authors’ response to R1 comment 5: Thank you for this constructive comment.

We agree that the elevated complication rates observed in nonagenarians undergoing cardiac surgery have significant implications for surgical decision-making in this cohort. Although advanced age alone should not preclude surgical intervention, the heightened risk of complications and mortality warrants careful consideration. Extant literature reports postoperative complication rates reaching 60% in nonagenarians undergoing major surgery, with a one-year mortality rate of approximately 20% following cardiac procedures.

We also agree that these substantial risks necessitate comprehensive discussions with patients and their families to facilitate informed decision-making. Surgeons must carefully balance the potential benefits of surgery against the associated risks, while considering patient preferences and anticipated postoperative quality of life. In assessing surgical candidacy, factors such as frailty, comorbidities, and functional status should be evaluated in conjunction with age. Moreover, the urgency of the procedure is a critical consideration, as elective surgeries generally yield more favorable outcomes compared to urgent or emergent cases. Ultimately, the decision to proceed with cardiac surgery in nonagenarians should be individualized, taking into account not only survival prospects but also the potential for improved functional status and quality of life.

Interestingly, among our major findings, we noted that we could find almost no published information that directly addresses this extremely important question. Although we collected some information on complication rates, which, intriguingly, appear to be somewhat lower than those cited for noncardiac surgery in this population, none of the studies reported patient-centred outcome measures. For example, we could find little to no information on patient days alive and out of the hospital, patient-reported outcome measures, or patient-reported experience measures.

We have taken your suggestion and added additional information and emphasis on this point to the statements regarding age and potential complication rates.

This has been updated in the discussion section under the section titled “Complications”.

6. Reviewer 1, comment 6: The manuscript identifies a lack of patient-reported outcomes but could suggest how future research might address this gap.

Authors’ response to R1 comment 6: Done. In the section addressing Future Directions, we have included a more extensive discussion of tools used to measure these outcomes that might be validated in this very elderly cohort (new references [71,72]).

7. Reviewer 1, comment 7: There are a few typographical errors and grammatical issues throughout the manuscript. A thorough proofreading would be beneficial. For example, the sentence "From the initial 1015 articles searched, we included twenty-eight studies from eight countries were included" is redundant and should be revised for clarity.

Authors’ response to R1 comment 7: The original manuscript has undergone thorough proofreading. All typographical errors and grammatical issues have been addressed, and additional changes have been made to improve clarity and readability throughout.

8. Reviewer 1, comment 8: Ethical Considerations: A discussion of the ethical implications of surgery in nonagenarians, particularly around informed consent, would be valuable.

Authors’ response to R1 comment 8: Done. We have highlighted this specific concern in the Discussion section (with new references [45,46]).

Reviewer TWO

We thank Reviewer Two for their expert insights and time spent reviewing our manuscript. We have provided a detailed comment to each of the reviewer’s questions, as outlined below.

1. Reviewer 2, Comment 1: The authors have to make a choice between a systematic and a scoping review that they seem to assimilate (even in title) but which are two different type of studies with different aim and methodology. See Munn, Z., Peters, M.D.J., Stern, C. et al. Systematic review or scoping review? Guidance for authors when choosing between a systematic or scoping review approach. BMC Med Res Methodol 18, 143 (2018). https://doi.org/10.1186/s12874-018-0611-x.

Authors’ response to R2 comment 1: Thank you for this important question. We appreciate the opportunity to clarify our rationale for conducting a scoping review rather than a systematic review.

The primary reason for choosing a scoping review methodology is the current state of the literature in this area. Given the relatively sparse and heterogeneous body of research regarding the outcomes of cardiac surgery in nonagenarians, the use of the scoping review format permitted us to generate a comprehensive map of all available evidence, identify key themes, and highlight gaps in the literature. Unlike a systematic review, which focuses on synthesizing high-quality studies to answer a specific research question, a scoping review provides a broader overview of the extent, range, and nature of the existing research.

Additionally, a scoping review is particularly suited for an emerging or complex area such as this, where variations in study design, outcome measures, and patient cohorts make direct comparison difficult. By capturing a wide spectrum of evidence, including retrospective cohort studies, case series, and observational studies, we can better understand the breadth of available data and assess whether a more targeted systematic review would be feasible in the future. Furthermore, our approach aligns with best practices for evidence synthesis in areas with limited randomized controlled trials or standardized reporting. This methodology enables us to categorize different study types, methodologies, and reported outcomes without the stringent inclusion criteria that a systematic review would necessitate.

In summary, our decision to conduct a scoping review stems from the need to systematically explore and describe the literature landscape in this field. We believe this approach has provided valuable insights into the current knowledge base that

---

## [Decision Letter · Decision Letter 1]

23 Jun 2025

Dear Dr. Weinberg,

Thank you for submitting your manuscript to PLOS ONE. After careful consideration, we feel that it has merit but does not fully meet PLOS ONE’s publication criteria as it currently stands. Therefore, we invite you to submit a revised version of the manuscript that addresses the points raised during the review process.

The manuscript has been evaluated by two reviewers, and their comments are available below.

The reviewers have raised a number of concerns. They feel the manuscript should be clearer in regards to the methodology used for data aggregation and charting and any limitations thereof, especially as it relates to estimates of mortality rate. The specific section on this (lines 349-369) could benefit from a discussion of these limitations. This is particularly important given the large variability in the number of patients in the studies included in your review. Additionally, the reviewers raised issues with the thematic analysis of patient-reported outcome measures and quality-of-life measures. I understand these were not carried out due to lack of data, but I think the text could be clarified regarding this point (especially e.g. lines 211-217) with an acknowledgement that this was a planned analysis that could not in the end be done.

Could you please carefully revise the manuscript to address all comments raised?

We look forward to receiving your revised manuscript.

Kind regards,

Alejandro Torrado Pacheco, Ph.D.

Associated Editor

PLOS One

Reviewers' comments:

Reviewer's Responses to Questions

**Comments to the Author**

Reviewer #1: All comments have been addressed

Reviewer #4: All comments have been addressed

2. Is the manuscript technically sound, and do the data support the conclusions?

Reviewer #1: Yes

Reviewer #4: Partly

3. Has the statistical analysis been performed appropriately and rigorously?

Reviewer #1: Yes

Reviewer #4: No

4. Have the authors made all data underlying the findings in their manuscript fully available?

Reviewer #1: Yes

Reviewer #4: Yes

5. Is the manuscript presented in an intelligible fashion and written in standard English?

Reviewer #1: Yes

Reviewer #4: Yes

Reviewer #1: I appreciate the authors' thorough engagement with all feedback. The manuscript now appears well-prepared.

Reviewer #4: some methodological issues, highlighted in attached document, overall very interesting topic and merits publication with minor adjustments.

**Do you want your identity to be public for this peer review?** For information about this choice, including consent withdrawal, please see our Privacy Policy

Reviewer #1: **Yes: ** Yousef Tanas

Reviewer #4: No

---

## [Author Response · Author response to Decision Letter 2]

9 Jul 2025

Professor Alejandro Torrado Pacheco, Ph.D.

Associated Editor

PLOS One

9th July 2025

Ref: Second Resubmission of PONE-D-24-26035

Title: Outcomes and complications in nonagenarians undergoing cardiac surgery: a scoping review

Dear Dr. Torrado Pacheco

Once again, my coauthors and I would like to thank you and the expert reviewers for taking the time to provide a very constructive and thoughtful review of our manuscript.

As requested, we have included the following items with our resubmission:

1. A point-by-point response letter (to follow) that addresses each issue raised by the Academic Editor and the Reviewers. I have uploaded this letter as a separate file labelled “Response to Reviewers.”

2. A marked-up copy of the manuscript that highlights in RED changes made to the original version. This has been uploaded as a separate file labelled “Revised Manuscript with Tracked Changes.”

3. An unmarked version of our revised paper without tracked changes. This has also been uploaded as a separate file labelled “Manuscript.”

Comments from the Expert REVIEWERS

Reviewer 1: I appreciate the authors' thorough engagement with all feedback. The manuscript now appears well-prepared.

Authors’ response: We sincerely appreciate the reviewer's recognition of this important research gap. We thank the Reviewer for their take taking to review the resubmission and for the constructive comment above.

Reviewer 4: Some methodological issues, highlighted in attached document, overall very interesting topic and merits publication with minor adjustments.

Authors’ response: Thank you for taking the time to review our manuscript and for providing additional comment. We appreciate the opportunity to further revise our manuscript and enhance the scientific merits of the paper. Please find below a detailed response to each of your comments

Reviewer Q1. Relevant topic, definite gap in the literature with regard to surgery on nonagenarians

Authors’ response to Q1: We sincerely appreciate the reviewer's recognition of this important research gap. The paucity of evidence regarding surgical outcomes in nonagenarians represents a critical knowledge deficit in contemporary cardiac surgery practice. With the global nonagenarian population projected to exceed 30 million by 2030, and increasing prevalence of cardiovascular disease in this demographic, establishing evidence-based outcomes data is essential for informed clinical decision-making.

Our scoping review methodology was specifically chosen to systematically map the available evidence landscape, identify knowledge gaps, and provide a foundation for future research priorities in this rapidly expanding patient population.

Reviewer Q2. Most points have been covered by the preceding 3 reviewers, I only have a couple of additions myself

Authors’ response to Q2: We acknowledge the reviewer's collaborative approach and appreciate the opportunity to address additional methodological considerations that complement the feedback from previous reviewers. This iterative peer review process strengthens the scientific rigor and clinical relevance of our work.

Reviewer Q3. Not sure why a pilot review was necessary, it seems like it was actually the initial step of the review and not a separate or discovery mission

Authors’ response to Q3: Thank you for this important comment that we have discussed with our statistician who is also a co-author on this paper. In hindsight, we agree that the pilot review may not have been necessary. We were however unsure how many papers would need to be screened, and we felt that the pilot review was a standard methodological step to calibrate inter-rater consistency and ensure that inclusion/exclusion criteria were applied consistently before full screening.

We were also cognizant that the pilot review was implemented as a methodological quality assurance measure consistent with established scoping review guidelines. According to the JBI Manual for Evidence Synthesis and PRISMA-ScR recommendations, pilot testing serves multiple critical functions: (1) calibrating reviewer agreement and ensuring consistent application of inclusion/exclusion criteria, (2) validating data extraction instruments, and (3) identifying potential methodological challenges before full-scale implementation.

As described in the Methods, “Study selection and screening procedure”, this step helped refine our screening tool, improve reviewer agreement, and ensured methodological transparency consistent with PRISMA-ScR and Levac et al.’s framework. Our pilot demonstrated substantial inter-rater reliability (Cohen's kappa = 0.643, 95.6% concordance), which provided methodological confidence for the complete review. Whilst this approach is not always routine in scoping reviews it enabled a robust and systematic synthesis of the evidence and ensured reproducibility and reliability of findings.

Reviewer Q4. I do agree with one of the reviews that there is some overlap between scoping and systematic reviews in this study, the study should, strictly speaking, report on the available literature, without seeking to synthesise it as that is the objective of a systematic review, as stated by the authors in line 147. for example, it is difficult to understand how they came up with the mortality rate of 10.5% when there is so much heterogeneity between the studies, with some studies having as small a sample of 9 and others as large a sample as over 2000, adding means from such differently powered studies may give an inaccurate result without weighting, which cannot be done for heterogenous studies.

Authors’ response to Q4: We acknowledge the reviewer's important observation regarding methodological boundaries between scoping and systematic reviews. However, our approach aligns with established scoping review methodology as outlined in the PRISMA-ScR guidelines. Scoping reviews legitimately provide descriptive numerical summaries of available evidence without formal meta-analytical synthesis. The reported median mortality rate of 10.5% (IQR 7.2–14.6%) represents a descriptive statistic of the available evidence landscape, not a weighted pooled meta-analytical estimate. This has been added in to the Results sections (Lines 230-232)

We calculated medians and interquartile ranges precisely because of the heterogeneity you identify-this approach is less sensitive to extreme values and varying sample sizes than weighted means would be. The primary objective was to map the breadth of available evidence and identify patterns, which differs fundamentally from systematic review synthesis that would require formal quality assessment and meta-analytical techniques with appropriate weighting strategies.

Given this important comment, we have amended the Results and Discussion sections to more explicitly clarify that our findings are descriptive and derived from unweighted medians rather than means or pooled statistics, to avoid overinterpretation.

Reviewer Q5. It is interesting to see the huge discrepancy in reported mortality rate in nonagenarians undergoing open heart surgery (10.5%) compared to nonagenarians undergoing non cardiac surgery (38%), It is difficult to believe that there is a reversal in the mortality rate in non-cardiac surgery compared to open cardiac surgery observed in the general population compared to that in such a high risk population. After all, a recent study in Italy identified the mortality in non-surgical nonagenarians to be 20.3% (Pancani, S., Lombardi, G., Sofi, F. et al. 12-month survival in nonagenarians inside the Mugello study: on the way to live a century. BMC Geriatr 22, 194 (2022). https://doi.org/10.1186/s12877-022-02908-9). While the authors ascribe it to careful patient selection in cardiac surgery, could other factors have contributed, such as the statistical analysis problems mentioned above? Or if indeed this is a true result, could it be that this represents the findings in first-world countries, as most of the data was obtained from high-income countries?

Authors’ response to Q5: The reviewer raises an astute observation about the apparent mortality paradox. We have included an additional paragraph in the discussion section that provided deeper insights. In the discussion section (Lines 357-369) we now state:

“Several factors beyond the statistical methods are likely to have contributed to this counterintuitive finding. First, the stringent patient selection for cardiac surgery in nonagenarians creates a highly selected cohort: patients must demonstrate adequate physiological reserves, cognitive function, and life expectancy to justify the procedural risk. In contrast, non-cardiac surgery populations might include emergent presentations with less rigorous preoperative optimization. Second, the availability of transcatheter alternatives might selectively direct higher-risk nonagenarians away from open surgery. Frail patients might be triaged toward transcatheter aortic valve replacement or medical management, thus further contributing to the lower observed mortality. Third, the reported increase in mortality of nonagenarians undergoing non-cardiac surgery represents a general nonagenarian population rather than a highly selected cardiac-surgery cohort. Fourth, the data reflect predominantly high-income countries with advanced perioperative care systems, specialized cardiac surgery centers, and comprehensive postoperative support.

In the limitations section (Lines 513-515), we have also added a cautionary note about generalizing these results to lower-resource settings where risk profiles and outcomes may differ significantly. We now state “Finally, our findings cannot be generalized to lower-resource settings or lower- and middle-income countries, where risk profiles and outcomes may differ significantly.”

Reviewer Q6. No mention as to whether the procedures documented were performed as elective or emergency procedures as the mortality rate may differ significantly between elective and emergency cardiac surgery

Authors’ response to Q6: Thank you for raising this excellent point. We have reviewed each included paper once again to see if this was documented or captured in the original publication. Unfortunately, most included studies did not stratify outcomes based on surgical urgency.

We acknowledge this limitation (See lines 489-491), where we now state: “Unfortunately, most included studies did not stratify outcomes by surgical urgency, thus constraining our ability to evaluate the differential effects of elective vs. emergent procedures.” This omission constrains our ability to evaluate the differential impact of elective vs. emergent procedures. We have added this limitation to the Discussion as above, and updated Table 3 to recommend that future studies report surgical urgency as a key preoperative variable for risk stratification.

Reviewer Q7. Scoping review objectives differ between the introduction and the objectives, could do with some clarity, the third objective (Proms) was not met as there was insufficient data

Authors’ response to Q7: Thank you for highlighting this discrepancy. We acknowledge the need for clarity between our introduction and formal objectives.

The apparent discrepancy reflects the evolution from our initial comprehensive research questions to the practical limitations encountered during evidence mapping. While we initially aimed to evaluate PROMs and PREMs comprehensively, the complete absence of such data in the literature necessitated reframing this as a critical gap identification rather than a quantitative analysis.

Both PREMS and PROMS represents legitimate scoping review outcomes-identifying what evidence exists and, equally importantly, what evidence is absent. We have clarified our objectives to reflect that PROMs/PREMs assessment was exploratory, with the understanding that absence of such data would itself constitute a significant finding. While PROMs and PREMs were included as objectives to identify literature gaps, we now explicitly state that no data were available, thus rendering that objective unmet. This reinforces the central finding of our review: a significant absence of patient-centered outcomes in this demographic.

We have revised the Introduction and Methods to ensure alignment with the final objectives stated.

• In the methods section (Lines 89-93) we state: “Finally, as an exploratory outcome, the review identified whether patient-reported outcomes (PROMs) and experience measures (PREMs) have been included in studies examining outcomes in nonagenarians undergoing cardiac surgery, given that an absence of such data would itself constitute a notable finding.”

• In the data synthesis and analysis section (Lines 211-213) we state: “Our approach can therefore be accurately described as a narrative synthesis of quantitative findings, organized thematically around clinical domains (mortality, morbidity, and length of stay).”

• In the section Patient-reported experience and outcomes measures section (Lines 320-321) we state: “None of the included studies reported PROMs or PREMs; therefore, this study objective was unmet.”

Reviewer Q8. According to the comments made in response to previous reviewer comments, no data could be found regarding patient satisfaction measures, yet it is mentioned in the objectives and methods under data extraction, and discussion see lines 92, 194, 195,204, 211,212 and 331 etc. Should this section then not be removed from the objectives and methods, and a comment made only in the discussion, identifying it as a gap in the literature?

Authors’ response to Q8: The reviewer correctly identifies an apparent inconsistency in our methodology. This has been corrected, and this section has been removed from the objectives and methods.

• The methods section (Lines 89-93) states: “Finally, as an exploratory outcome, the review identified whether patient-reported outcomes (PROMs) and experience measures (PREMs) have been included in studies examining outcomes in nonagenarians undergoing cardiac surgery, given that an absence of such data would itself constitute a notable finding.”

The methodology sections appropriately described our comprehensive search strategy, while the results and discussion sections report the significant finding that no such data exists. This absence of patient-centered outcome measures represents a critical knowledge gap that our review identified.

Reviewer Q9. Again in the data analysis section we find some overlap in terms of methodology, thematic analysis is a qualitative analytic tool and depends on the conceptual framework and selection of appropriate paradigm, the context of which is of the utmost importance, this makes it technically difficult to code and draw themes from an analysis of retrospective quantitative data obtained from thousands of patients who had different research tools used in many varying countries with differing languages, especially as no patient satisfaction measures were identified in the data, the appropriate analysis of such data in the case of a scoping review is simply narrative

Authors’ response to Q9: The reviewer raises important considerations about thematic analysis methodology. We acknowledge that our use of the term "thematic analysis" may be imprecise in this context. Given the absence of qualitative data or patient-reported outcomes in the included studies, formal thematic analysis was indeed impossible. Our approach is more accurately described as narrative synthesis of quantitative findings, organized thematically around clinical domains (mortality, morbidity, length of stay).

We have revised our methodology description to more accurately reflect this narrative approach rather than formal thematic analysis, which requires qualitative data and interpretive frameworks as the reviewer correctly notes. Given the paucity of such data, we have removed thematic analysis from our stated analytical methods and revised the Data Analysis section to reflect a narrative synthesis alone. These changes help ensure methodological consistency and prevent overreach.

Thank you for this important suggestion.

Reviewer Q10. According to the authors, the majority

---

## [Decision Letter · Decision Letter 2]

21 Aug 2025

Outcomes and complications among nonagenarians undergoing cardiac surgery: a scoping review

PONE-D-24-26035R2

Dear Dr. Weinberg,

We’re pleased to inform you that your manuscript has been judged scientifically suitable for publication and will be formally accepted for publication once it meets all outstanding technical requirements.

Kind regards,

Dr Redoy Ranjan, MBBS, MRCSEd, Ch.M., MS (CV&TS), FACS

Academic Editor

PLOS ONE

Additional Editor Comments (optional):

Reviewers' comments:

Reviewer's Responses to Questions

**Comments to the Author**

Reviewer #4: All comments have been addressed

2. Is the manuscript technically sound, and do the data support the conclusions?

Reviewer #4: Yes

3. Has the statistical analysis been performed appropriately and rigorously?

Reviewer #4: Yes

4. Have the authors made all data underlying the findings in their manuscript fully available?

Reviewer #4: Yes

5. Is the manuscript presented in an intelligible fashion and written in standard English?

Reviewer #4: Yes

Reviewer #4: (No Response)

**Do you want your identity to be public for this peer review?** For information about this choice, including consent withdrawal, please see our Privacy Policy

Reviewer #4: **Yes: ** Mulai Slave

---

## [Editor Report · Acceptance letter]

PONE-D-24-26035R2

PLOS ONE

Dear Dr. Weinberg,

I'm pleased to inform you that your manuscript has been deemed suitable for publication in PLOS ONE. Congratulations! Your manuscript is now being handed over to our production team.

Kind regards,

on behalf of

Dr. Redoy Ranjan

Academic Editor

PLOS ONE